# Probiotic-Treated Super-Charged NK Cells Efficiently Clear Poorly Differentiated Pancreatic Tumors in Hu-BLT Mice

**DOI:** 10.3390/cancers12010063

**Published:** 2019-12-24

**Authors:** Kawaljit Kaur, Anna Karolina Kozlowska, Paytsar Topchyan, Meng-Wei Ko, Nick Ohanian, Jessica Chiang, Jessica Cook, Phyu Ou Maung, So-Hyun Park, Nicholas Cacalano, Changge Fang, Anahid Jewett

**Affiliations:** 1Division of Oral Biology and Medicine, The Jane and Jerry Weintraub Center for Reconstructive Biotechnology, Department of Dentistry, University of California Los Angeles (UCLA), Los Angeles, CA 90095, USA; drkawalmann@gmail.com (K.K.); kozlowa@ump.edu.pl (A.K.K.); paytsss20@gmail.com (P.T.); mengwei@ucla.edu (M.-W.K.); nickohanian@g.ucla.edu (N.O.); jesswchiang@ucla.edu (J.C.); jessicook18@ucla.edu (J.C.); phyuou.maung@gmail.com (P.O.M.); nefertiti8625@gmail.com (S.-H.P.); 2Department of Tumor Immunology, Chair of Medical Biotechnology, Poznan University of Medical Sciences, 61-701 Poznan, Poland; 3The Jonsson Comprehensive Cancer Center, Los Angeles, CA 90095, USA; ncacalan@ucla.edu; 4Department of Radiation Oncology, Division of Molecular and Cellular Oncology, UCLA School of Dentistry and Medicine, Los Angeles, CA 90095, USA; 5BioPro Diagnostics, LLC, 4919 Brook Hills Drive, Annandale, VA 22003, USA; changgefang@hotmail.com

**Keywords:** IFN-γ, NK cells, cytotoxicity, probiotics, stem-like/poorly differentiated pancreatic cancer, differentiation, hu-BLT mice, super-charged NK cells

## Abstract

*Background and Aims:* We have previously demonstrated that the stage of differentiation of tumors has profound effect on the function of NK cells, and that stem-like/poorly differentiated tumors were preferentially targeted by the NK cells. Therefore, in this study we determined the role of super-charged NK cells in immune mobilization, lysis, and differentiation of stem-like/undifferentiated tumors implanted in the pancreas of humanized-BLT (hu-BLT) mice fed with or without AJ2 probiotics. The phenotype, growth rate and metastatic potential of pancreatic tumors differentiated by the NK cells (NK-differentiated) or patient derived differentiated or stem-like/undifferentiated pancreatic tumors were investigated. *Methods:* Pancreatic tumor implantation was performed in NSG and hu-BLT mice. Stage of differentiation of tumors was determined using our published criteria for well-differentiated tumors exhibiting higher surface expression of MHC- class I, CD54, and PD-L1 (B7H1) and lower expression of CD44 receptors. The inverse was seen for poorly-differentiated tumors. *Results:* Stem-like/undifferentiated pancreatic tumors grew rapidly and formed large tumors and exhibited lower expression of above-mentioned differentiation antigens in the pancreas of NSG and hu-BLT mice. Unlike stem-like/undifferentiated tumors, NK-differentiated MP2 (MiaPaCa-2) tumors or patient-derived differentiated tumors were not able to grow or grew smaller tumors, and were unable to metastasize in NSG or hu-BLT mice, and they were susceptible to chemotherapeutic drugs. Stem-like/undifferentiated pancreatic tumors implanted in the pancreas of hu-BLT mice and injected with super-charged NK cells formed much smaller tumors, proliferated less, and exhibited differentiated phenotype. When differentiation of stem-like tumors by the NK cells was prevented by the addition of antibodies to IFN-γ and TNF-α, tumors grew rapidly and metastasized, and they remained resistant to chemotherapeutic drugs. Greater numbers of immune cells infiltrated the tumors of NK-injected and AJ2-probiotic bacteria-fed mice. Moreover, increased IFN-γ secretion in the presence of decreased IL-6 was seen in tumors resected and cultured from NK-injected and AJ2 fed mice. Tumor-induced decreases in NK cytotoxicity and IFN-γ secretion were restored/increased within PBMCs, spleen, and bone marrow when mice received NK cells and were fed with AJ2. *Conclusion:* NK cells prevent growth of pancreatic tumors through lysis and differentiation, thereby curtailing the growth and metastatic potential of stem-like/undifferentiated-tumors.

## 1. Introduction

Despite improvements in the therapeutic strategies, the five-year survival rate for pancreatic cancer patients remains dismal at 5%, with limited effectiveness of chemotherapeutic drugs in targeting these tumors [1,2,3]. In previous studies, CD44+CD24+ESA+ and CD44+/CD133+/EpCAM+ triple positive MiaPaCa-2 (MP2) pancreatic cancer stem cells (CSCs) [4,5,6] were shown to have increased cell growth, migration, clonogenicity, self-renewal capacity, and chemotherapy resistance, and, therefore, formed one of the reasons for the selection of MP2 tumors in our studies [7,8,9].

It has previously been reported that decreased expression of MHC-class I is a hallmark of poorly differentiated tumors [10,11]. Therefore, low MHC-class I expression might favor survival of CSCs and explain the limited effectiveness of T-cell based immunotherapies in cancer patients [12,13]. However, we have previously shown that CSCs are excellent targets for NK cell-mediated cytotoxicity, whereas their differentiated counterparts are significantly more resistant [14]. Furthermore, we have also reported that de-differentiation of tumors resulted in increased NK cell-mediated cytotoxicity [15,16,17]. Thus, the stage of differentiation of tumors is a predictor of immunosuppression and can be used to assess the prognosis of cancer [18,19,20,21]. Higher expression of MHC-class I and MHC-class II in well differentiated tumors was correlated with increased numbers of tumor infiltrating immune cells, and was associated with better prognosis in cancer patients [18,21]. In contrast, higher expression of CD44 on the tumor cells [20] and increased PD-1 expression in the peripheral blood of patients [19] was correlated with poor tumor differentiation and the aggressiveness of the tumors. In agreement, we have previously shown that increased MHC-class I, CD54, and PD-L1 expression on tumors is correlated with well-differentiated tumors and better prognosis in our in vitro and in vivo studies [22,23].

It is known that cytotoxic function of NK cells is suppressed after their interaction with CSCs/stem cells [24,25,26]. We therefore proposed, and recently demonstrated that NK cells, as a result of interaction with CSCs, undergo split-anergy, a key event in which NK cytotoxicity is lost but a greater secretion of IFN-γ ensues, promoting an increase in the differentiation antigen expression of MHC-class I, CD54, and PD-L1 on CSCs [22,27] which has recently been shown to co-relate with effectiveness of anti-PD-1 therapy [28,29]. Indeed, increased numbers of circulating NK cells have been correlated with better prognosis [30,31]. However, NK cell cytotoxic activity in cancer patients is severely reduced, correlating with the decreased expression of NK cell activating receptors even at the early stages of disease, and are further diminished in advanced cancers [32,33,34].

Patients’ NK cells are significantly defective in their function, and the defect is seen at both pre-neoplastic and neoplastic stages of pancreatic cancer [35,36,37]. Patients’ NK cells do not recover their full functional potential even when the best activating conditions are provided for their expansion and function [35]. Moreover, patients’ NK cells under the expansion conditions give rise to T cell expansion at a much faster rate, with a subsequent decrease in the percentages of NK cells when compared to those from healthy individuals [35]. Therefore, the use of allogeneic NK cells from healthy individuals is preferable than autologous NK cells for therapeutic purposes. The use of allogeneic NK cells in hematologic malignancies following HLA-haploidentical HSCT in clinical trials has previously been reported [38,39]. In addition, allogeneic NK cells exert less collateral damage due to graft versus host disease (GVHD) when used therapeutically in solid tumors [40,41,42,43]. In search of a more potent therapeutic dose of NK cells, we have recently established a novel strategy to expand highly functional NK cells, coined as super-charged NK cells, by employing osteoclasts as feeder cells in the presence of a combination of sonicated probiotic bacteria (sAJ2) [27]. The potency and effectiveness of super-charged NK cells are significantly superior to those established by other methodologies or when compared to primary activated NK cells [35]. The term super-charged was used to describe the magnitude and superiority of their functional potential in lysing and differentiating CSCs/poorly differentiated tumors [35].

In close agreement with our in vitro studies, we show in this report the direct correlation between increased MHC-class I, CD54, and PD-L1 expression on tumor cells and degree of differentiation and better prognosis for pancreatic tumors in in vivo experiments. Therefore, in mouse models of NSG and hu-BLT, we show that NK cells drive selection and differentiation of pancreatic CSCs/poorly differentiated tumors, resulting in inhibition of their aggressiveness, metastatic potential, and increased susceptibility to chemotherapy. The in vivo data presented in this paper largely complement our in vitro findings reported previously for the role of NK cells in the selection and differentiation of poorly differentiated tumors of glioblastoma, pancreatic, oral, lung, breast, and melanomas [44]. Based on the data presented here, we propose that in vivo selection and differentiation of poorly differentiated tumors by functionally competent NK cells, which are lacking in cancer patients, is a key event in reducing the aggressiveness of these tumors.

## 2. Results

### 2.1. Correlation between NK Cell Cytotoxicity and the Stage of Differentiation in Pancreatic Tumors

Six different pancreatic tumor cell lines each characterized at poorly, intermediate, and well differentiated stages pathologically by other laboratories previously [45] were used to determine phenotype, susceptibility to NK cell-mediated cytotoxicity and secretion of IFN-γ directly correlating with the differentiation stages of the tumors. Poorly differentiated MP2 and Panc-1 demonstrated moderate to low levels of MHC-class I and CD54 in the presence of higher surface expression of CD44 receptors. Moderately differentiated BXPC3 and HPAF exhibited higher levels of MHC-class I surface expression in the presence of moderate to high expression of surface CD44 and CD54 receptors, and well-differentiated Capan and PL12 expressed higher levels of surface CD54 and MHC-class I in the presence of lower CD44 surface expression (Figure 1A). Furthermore, the stage of differentiation of the tumors was correlated with sensitivity to NK cell mediated cytotoxicity in pancreatic tumor cells. The highest susceptibility to NK cell mediated cytotoxicity was seen with undifferentiated MP2 and Panc-1 tumors; whereas the well differentiated PL12 and Capan tumors demonstrated the lowest sensitivity to NK mediated lysis (Figure 1B and Appendix A). BXPC3 and HPAF, being moderately differentiated tumors, exhibited intermediate sensitivity to NK cell lysis (Figure 1B). Therefore, a direct correlation between augmented sensitivity to NK mediated lysis and poor differentiation of pancreatic tumors was evident from these experiments. Identical results in tumor differentiation were seen in oral and glioblastoma tumors [44,46], and more recently in melanoma and lung tumors (manuscript in prep and submitted and [47,48]). Similar to PL12 and Capan, NK-differentiated MP2 tumors exhibited identical surface receptor phenotype, and were resistant to NK cell mediated cytotoxicity [44]. NK cells induced differentiation of MP2 tumors through the functions of IFN-γ and TNF-α, with rhIFN-γ and/or rhTNF-α exhibiting similar results to NK-induced IFN-γ and TNF-α [22] (see Appendix A of the results and Appendix A in Appendix A) [27].

### 2.2. Curtailed Pancreatic Tumor Growth and Long-Term Survival of Mice after Implantation of NK-Differentiated MP2 and Patient-Derived Differentiated PL12 Tumors

MP2 tumors (3 × 10^5^) implanted in the pancreas of NSG mice grew within 4 weeks and metastasized to the liver and caused significant morbidity and mortality in the mice (Figure 2 and Appendix A), whereas mice injected with greater numbers of PL12 tumors (2 × 10^6^) generated no or very small tumors within 12 weeks and the tumors did not metastasize nor caused morbidity in the mice (Figure 2 and Appendix A). Injection of NK-differentiated MP2 tumors (5 × 10^5^) to pancreas of NSG mice neither exhibited visible tumor growth nor tumors metastasized to the liver, and all mice survived at 12 weeks when the experiments were terminated (Figure 2 and Appendix A).

### 2.3. NK-Differentiated MP2 Tumors Did Not Grow Visible Tumors in the Pancreas of Hu-BLT Mice

Hu-BLT mice were generated (Appendix A), and the successful reconstitution of human immune cells in spleen, bone marrow, and peripheral blood (Appendix A) were verified, and the levels of different immune subsets in peripheral blood (Appendix A) and pancreas (Appendix A) were determined, and the results were compared to peripheral blood from human donors (Appendix A). Hu-BLT NK cells purified from the spleen of mice responded to the activation signals provided by the IL-2 and anti-CD16 mAb treatment and expanded greatly, and demonstrated increased secretion of IFN-γ when cultured with both autologous and allogeneic osteoclasts in the presence of sAJ2 treatment (Appendix A), indicating close similarity between hu-BLT and human donor derived NK cell expansion and function by osteoclasts. Therefore, although the frequencies of NK cells are lower in the peripheral blood of hu-BLT mice, their function is similar to those obtained from human donors. Hu-BLT mice were implanted with undifferentiated MP2 tumors (Figure 3A) and those differentiated with NK-supernatants as described before [22,27,49] (Appendix A) in the pancreas, and their growth dynamics and overall effect on mice were studied. MP2 tumors grew rapidly and formed tumors in the pancreas, and mice exhibited all the signs of morbidity within 6–7 weeks, and upon sacrifice at week 7, they exhibited tumors which spanned the entire abdomen and enveloped the spleen, stomach, and a portion of intestines (Figure 3B, panel a). When NK-differentiated MP2 tumors were implanted in mice, no tumors were seen, and mice did not exhibit any signs of morbidity (Figure 3B, panel c). In in vitro cell cultures, NK-differentiated MP2 tumors similar to patient derived PL12 differentiated tumors grew slower when compared to undifferentiated MP2 tumors [44]. The proportions of huCD45+ cells in pancreas were significantly decreased in mice implanted with MP2 tumors (3.37%) when compared to control mice (7.46%) likely reflecting the increased tumor burden in these mice (Appendix A), however, those implanted with NK-differentiated MP2 tumors maintained higher proportions of huCD45+ cells (10.19%), and furthermore, the percentages of huCD3+ T cells within huCD45+ cells were much higher in MP2 implanted tumors (80%) when compared to either NK-differentiated MP2 tumor implanted mice (62%) or control mice (45%) (Figure 3C and Appendix A).

### 2.4. Single Injection of NK Cells Inhibited Tumor Growth in Mice Implanted with MP2 Tumors

Mice implanted with MP2 tumors and injected with 1.5 × 10^6^ super-charged NK cells with potent cytotoxic and cytokine secretion capabilities (Figure 3A) [35] exhibited no or substantially smaller tumors, without the involvement of other organs or signs of morbidity (Figure 3B, panel b). Due to increased morbidity and mortality in tumor bearing mice by fast growing tumors in 7 weeks after tumor implantation, we shortened the time of sacrifice to 4–5 weeks after tumor implantation to be able to study the pancreas and the dynamics of immune cell infiltration in the pancreas. Greater numbers of huCD45+ cells were seen in the single cells prepared from the dissociated pancreas of either NK-injected tumor-bearing mice (9.2%) or NK-differentiated tumor-implanted mice (10.19%) or in the healthy control mice (7.46%) when compared to those from tumor-bearing mice (3.37%) (Appendix A). In addition, increased percentages of huCD45+CD3+ T cells were seen in cells dissociated from the pancreas of MP2 implanted mice (Figure 3C and Appendix A), whereas cells dissociated from the pancreas of tumor-bearing mice which received NK cells or from mice with implanted NK-differentiated tumors or from control healthy mice exhibited relatively lower percentages of huCD45+CD3+ T cells (Figure 3C and Appendix A) and higher percentages of NK cells in the dissociated pancreas when compared to those from tumor-bearing mice (Figure 3D). Likewise, greater than two-fold increase in huCD16+ NK cells within the cells dissociated from the pancreas were seen from either NK-injected tumor-bearing mice or from the pancreas of healthy control mice with no tumor implantation, when compared to those from tumor-bearing mice (Figure 3D). 

Unlike tumor-bearing mice, when mice were fed AJ2 1–2 weeks before tumor implantation and injected with allogeneic or autologous super-charged NK cells (Figure 4A and Appendix A and please see below), their tumor weights remained substantially less (Figure 4B). No statistically significant differences in tumor weight could be observed between NK or NK injected/AJ2 fed mice, even though a slight decrease in the average tumor weight could be seen between the two groups (Figure 4B). This is likely due to the significant decrease already seen with NK injection alone in tumor bearing mice. Indeed, sera from the peripheral blood of either NK-injected or NK injected/AJ2 fed tumor-bearing mice exhibited 2.73 and 4.8-fold more IFN-γ, respectively, when compared to tumor-bearing mice (Figure 4C and Appendix A). Feeding AJ2 alone, or injecting super-charged NK cells in the absence of tumor implantation, or feeding AJ2 with implantation of tumors, or injecting super-charged NK cells and feeding AJ2 all increased the levels of IFN-γ in the serum of the hu-BLT mice moderately when compared to control mice in the absence of any treatments. These mice had much less IFN-γ in the sera when compared to those which were implanted with the tumor and fed with AJ2 and injected with super-charged NK cells (Appendix A). Mice with implantation of the tumor in the absence of any treatment had the least amount of IFN-γ in the sera (Figure 4C and Appendix A). Similarly, mice implanted with NK-differentiated MP2 tumors (Appendix A) had minimal tumor weight (Figure 4D), and blocking MP2 differentiation with anti-IFN-γ and anti-TNF-α antibodies (Figure 4D and Appendix A), which we have described in the methods section in the current manuscript and in previous manuscripts [22,27], resulted in the inhibition of tumor differentiation and generation of tumors with higher tumor weights (Figure 4D).

When pancreata were removed, dissociated and equal numbers of cells were cultured from tumor-bearing mice which did not receive NK injection, attached colonies of tumors could be seen in 24–48 h and they grew rapidly thereafter, whereas those injected with allogeneic NK cells or autologous NK cells (Figure 4E,F) did not exhibit colonies initially, but a few were visible after day 5 or 6 and those colonies grew very slowly, and the numbers of tumors recovered remained substantially lower in comparison to those which did not receive NK injection (Figure 4E,F). Similarly, in NK-differentiated tumors, when implanted in mice and their pancreas were dissociated after sacrifice, tumors did not grow or grew very few colonies at later days and their growth remained extremely slow (Figure 4E,G), however, blocking differentiation with anti-IFN-γ and anti-TNF-α antibodies allowed attachment and growth of the tumors at 24–48 h with increased kinetics of growth (Figure 4G). Tumor growth after dissociation and plating was less in mice fed with AJ2 and injected with NK cells in comparison to NK alone injected mice, and both were substantially less than those which only received implantation of the MP2 tumors (Figure 4F). There was 18–22 fold more infiltrating huCD45+ immune cells in pancreas cultured from mice injected with tumors and NK cells in comparison to tumor-alone injected mice (Figure 4H,I and Appendix A). Greater percentages of infiltrating huCD45+ immune cells within the pancreas of NK injected tumor-bearing mice expressed CD94, and NKG2D surface receptors, whereas they expressed similar percentages of DNAM surface receptors when compared to tumor-bearing mice in the absence of NK injection (Figure 4H and Appendix A).

On average, a decrease in IFN-γ secretion from the pancreatic cell cultures could be observed in mice implanted with MP2 tumors, when compared to control mice with no tumors (Figure 4J). Injection of NK cells into tumor-bearing mice restored IFN-γ secretion in pancreatic cell cultures and the levels exceeded those seen in the control mice with no tumors (Figure 4J). Implantation of NK-differentiated MP2 tumors did not result in inhibition of IFN-γ in pancreatic cell cultures, and the amounts were comparable to those obtained from control mice with no tumors (Figure 4J). In contrast, IL-6 secretions were the highest in pancreatic cell cultures from tumor-bearing mice, and they were substantially lower in all other groups of mice (Figure 4K). Although slight differences could be seen between NK alone injected or NK-injected and AJ2 fed mice in terms of tumor weight/tumor growth in pancreas, there was, on average, higher secretion of IFN-γ by NK injected and AJ2 fed pancreatic cell cultures (Figure 4L).

MP2 tumors cultured from the pancreas of NK-injected mice exhibited increased expression of B7H1 (PD-L1), MHC-class I and CD54 when compared to tumor-bearing mice without NK injection (Figure 4M). Moreover, similar to our in vitro experiments, MP2 tumors cultured from the pancreas of NK-injected mice exhibited decreased sensitivity to NK cell-mediated lysis, whereas those cultured from tumor-bearing mice without NK injection exhibited increased susceptibility (Figure 4N,O) [44]). The addition of antibodies against IFN-γ and TNF-α during NK cell mediated differentiation of pancreatic tumors was found to restore the tumors’ susceptibility to NK cell-mediated cytotoxicity (Figure 4O). Since we were unable to recover enough tumors after implantation of NK-differentiated tumors in the pancreas; we were unable to run cytotoxicity assay.

### 2.5. Suppression of NK Cell Cytotoxicity and Decreased Secretion of IFN-γ in Tumor-Bearing Mice within All Tissue Compartments; Restoration by Super-Charged NK Cells

PBMCs from tumor-bearing mice (Figure 5A), which were similar to PBMCs (Appendix A) and NK cells (Appendix A) from pancreatic cancer patients, had significantly lower NK cell-mediated cytotoxicity and exhibited decreased IFN-γ secretion, when compared to those from healthy mice or humans, respectively. When PBMCs (Figure 5A–C), splenocytes (Figure 5D,F), enriched-NK cells from splenocytes (Figure 5G,H), huCD3+ T cells from splenocytes (Figure 5I), and BM-derived immune cells (Figure 5J–L) were assessed for NK cytotoxicity and/or secretion of IFN-γ, tumor-bearing mice had much lower cytotoxicity and/or secretion of IFN-γ in cells obtained from all tissue compartments, in comparison to those obtained from control mice without tumor, or tumor-bearing mice injected with NK cells, or those implanted with NK-differentiated tumors (Figure 5). Blocking NK differentiation of the tumors by anti-IFN-γ and anti-TNF-α antibodies resulted in a similar magnitude of IFN-γ secretion to those obtained from undifferentiated tumors in all tissue compartments tested (Figure 5C,F,L).

Similar to those seen with the pancreatic tumors, implantation of oral stem-like tumors in the oral cavity of hu-BLT mice resulted in similar profiles of cytotoxicity and secretion of IFN-γ from PBMCs isolated from oral tumor bearing mice in the presence and absence of NK injection (23). 

IV injection of anti-PD1 in combination with NK cells elevated secretion of IFN-γ in different tissue compartments tested (Appendix A). Anti-PD1 antibody injection in the absence of NK cells in tumor-bearing mice increased secretion of IFN-γ in the tissues (Appendix A).

### 2.6. Paclitaxel Induce Cell Death in NK-Differentiated MP2 Tumors Treated with/without N-acetyl Cysteine (NAC)

Unlike MP2 tumors, treatment of well-differentiated PL12 and Capan tumors with paclitaxel (Figure 6A) exhibited higher induction of cell death. Similarly, when MP2 tumors were differentiated with NK-supernatants and treated with paclitaxel, higher induction of cell death was observed in MP2 tumors (Figure 6B). Inhibition of NK-mediated differentiation by the addition of antibodies to IFN-γ and TNF-α substantially decreased the cell death induced by paclitaxel (Figure 6B). As shown in (Figure 6B), the addition of NAC to MP2, PL12, and Capan increased paclitaxel mediated cell death. Similarly, the addition of NAC to NK-supernatant differentiated MP2 tumors increased cell death, and blocking differentiation with IFN-γ and TNF-α mAbs decreased paclitaxel mediated cell death (Figure 6B). The differentiation potential of cells by NAC was shown before [50], and the addition of paclitaxel or cis-dichlorodiammineplatinum (CDDP or Cisplatin) to patient-derived differentiated oral squamous carcinoma cells (OSCCs) or NK-differentiated OSCSCs also mediated higher cell death, whereas minimal effects were seen on stem-like/poorly differentiated OSCSCs [44].

### 2.7. Monocytes or Osteoclasts from NK Injected Tumor Bearing Mice or NK-Differentiated Tumor Bearing Mice Had Higher Capacity to Activate NK Cells

When NK cells were cultured in the presence of autologous monocytes from tumor-bearing mice injected with the NK cells or those implanted with NK-differentiated MP2 tumors, they demonstrated increased secretion of IFN-γ (Figure 7A). Similarly, NK cells cultured with osteoclasts from tumor-bearing mice injected with NK cells or implanted with NK-differentiated tumors had significantly greater expansion and function of NK cells when compared to those from tumor-bearing mice in the absence of NK injection (Figure 7B–D). Similar results to those seen with tumor bearing hu-BLT mice were also seen when osteoclasts from pancreatic-cancer patients were cultured with NK cells (Appendix A). Osteoclasts from cancer patients were less able to expand NK cells (Appendix A), or increase NK cell-mediated cytotoxicity (Appendix A) or increase NK cell-mediated secretion of IFN-γ (Appendix A) when compared to those from healthy donors. When examining the surface receptor expression on cancer-patient and healthy individuals’ osteoclasts, decreased expression of MHC-class I, CD54, KLRG1, KIR2, and MICA/B could be seen on cancer patients’ OCs as compared to healthy OCs (Appendix A). 

Finally, when identical amounts of IFN-γ from the supernatants of NK cells were used to differentiate OSCSC tumors, those from pancreatic cancer patients’ NK cells were less effective in differentiating OSCSC tumors as compared to those from healthy donors’ NK cells (Appendix A). NK supernatants from patients elevated MHC-class I expression moderately (Appendix A) and induced only 35% resistance of OSCSC tumors to NK-mediated cytotoxicity (Appendix A), whereas NK supernatants from healthy individuals elevated MHC-class I substantially (Appendix A) and induced 78% resistance of OSCSCs against NK-mediated cytotoxicity (Appendix A). The rationale for using OSCSCs is because these cells are highly sensitive to IFN-γ mediated differentiation [14,49,51]. 

## 3. Discussion

NK cells limit growth and expansion of CSCs/poorly differentiated pancreatic tumors by tumor lysis and differentiation [22,27,44]. MP2 tumors, being poorly differentiated, form large tumors in NSG and hu-BLT mice, and have the ability to metastasize, whereas their NK-differentiated tumors or patient-derived well-differentiated tumors form very small tumors in the pancreas without metastatic potential. Indeed, the growth potential of MP2 tumors in in vitro cultures is found to be 10–15 fold, whereas those of the NK-differentiated counterparts are between 1.5–4 fold when the same numbers of tumors are cultured within the same time period [27,49,51], and no or slight cell death could be seen in the cultures of either undifferentiated MP2 tumors or those differentiated by the NK cells [27] (please see the Supplemental of Reference [27]). The slower growth rates of well differentiated pancreatic tumors in comparison to MP2 tumors were also shown previously [45].

Patient-derived PL12 tumors or NK-differentiated tumors, although not killed by primary NK cells, were however, susceptible to chemo-drugs and were killed by paclitaxel (Figure 6) as well as CDDP [44], whereas poorly differentiated tumors were resistant. Indeed, when NK-differentiation of MP2 tumors was inhibited by the combination of IFN-γ/TNF-α antibodies, tumors lost their sensitivity to chemotherapy and became susceptible to NK cell mediated cytotoxicity. Moreover, NAC, which is known to differentiate cells in addition to its other effects, [52] increased paclitaxel mediated death of NK-differentiated MP2 and well-differentiated tumors (Figure 6). 

Both autologous and allogeneic osteoclasts were able to expand hu-BLT NK cells with hu-BLT osteoclasts having slightly higher NK expansion potential and higher levels of IFN-γ secretion (Appendix A). Similarities in NK responses between hu-BLT and human NK cells to be expanded by their autologous osteoclasts, and secrete increased levels of IFN-γ and mediate augmented cytotoxicity partly provides the rationale for the use of this animal model as a surrogate model for the studies of human disease [35]. Furthermore, similar defects in both tumor-bearing hu-BLT and cancer patients’ NK cells were found. 

NK-differentiated MP2 tumors did not grow in hu-BLT mice, and when tumor differentiation was prevented by using antibodies to IFN-γ and TNF-α, tumors grew substantially (Figure 4G). In contrast, blocking IL-6 or IL-8 with antibodies was not able to influence differentiation of tumors by the NK cells [46]. Immunotherapy with super-charged NK cells in the presence or absence of AJ2 feeding resulted in a significant inhibition of tumor growth in hu-BLT mice. The rationale for feeding AJ2 was to maintain and increase NK cell activation *in vivo*, since recent studies from our laboratory and those of the others have shown significant increases in NK cell function by probiotic bacteria [27,53].

Tumors grew slower in tumor-bearing mice injected with NK cells, and they were of differentiated phenotype, whereas those in the absence of NK injection grew rapidly and remained undifferentiated. Moreover, tumors cultured from NK-injected tumor-bearing hu-BLT mice contained about 18–22 fold more huCD45+ immune cells and secreted higher IFN-γ in the presence of lower IL-6 secretion, whereas those cultured from tumor-bearing mice in the absence of NK injection had lower infiltrating huCD45+ cells and secreted lower IFN-γ in the presence of much higher IL-6 secretion. The increased secretion of IFN-γ was observed not only in tumor tissues, but also in all tissues examined from tumor-bearing mice fed with AJ2 and injected with NK cells when compared to those of tumor-bearing mice. Increased IL-6 secretion is likely due to the growing tumors in tumor-bearing mice [37].

The single injection of super-charged NK cells in tumor-bearing mice resulted in an increased surface receptor expression of PD-L1, CD54, and MHC-class I on tumor cells exhibiting decreased tumor growth and the loss of susceptibility of tumor cells to NK cell-mediated cytotoxicity (Figure 4L–O), potentially paving the road for their increased susceptibility to cytotoxic T lymphocyte (CTL) mediated killing due to increased MHC-class I expression. Increased percentages of T cells in the presence of decreased NK cells in the pancreas of tumor-bearing mice could be problematic for successful removal of undifferentiated tumors since these tumors are not eliminated by the T cells.

It should be emphasized that malignant tumors are not the only cells that are able to influence the function of NK cells within the tumor microenvironment. There are many other cells, including stromal cells such as tumor associated fibroblasts, fat cells, and other immune effectors within the pancreatic tumor microenvironment that could either increase the function of NK cells to drive differentiation of the tumors, or decrease their function resulting in the survival and expansion of stem-like/undifferentiated tumors depending on the early or late stages of cancer, respectively. In addition, at the later stages of cancer, many inhibitory effector cells such as T regulatory cells and MDSCs accumulate, and are therefore able to inhibit the function of NK cells resulting in the survival and expansion of cancer stem cells. Indeed, competent NK cells should be able to target and lyse MDSCs as they are able to lyse many different myeloid derived immune effectors.

In addition to releasing suppressive soluble factors into circulation, tumors can also suppress the function of NK cells by releasing various sized vesicles such as small, endosome-derived extracellular microvesicles of 30–100 nm exosomes which contain tumor proteins, mRNAs, and microRNAs, and larger-sized vesicles containing encapsulated cytosolic contents of 0.1 to 1 µm microparticles [54]. Thus, tumors can profoundly inhibit the function of NK cells in cancer patients locally within the tumor microenvironment, and distantly within the peripheral blood and healthy tissues leading to irreversible damage of patients’ NK cells.

Similar to cancer patients’ monocytes and osteoclasts, those from tumor-bearing mice had much lower ability to expand autologous or allogeneic NK cells or increase their functional potential. More severe inhibition of NK cell expansion and function is seen when both NK and monocytes are from tumor-bearing mice due to the combined defects in both NK cells and monocytes. These experiments not only highlight similarities between the tumor-bearing hu-BLT mouse model and human cancers, but also indicate a severe functional deficiency in NK cell activating effectors in tumor-bearing hu-BLT mice similar to cancer patients. It is also important to note that the highest activation of NK cells from hu-BLT mice was achieved through the implantation of NK-differentiated tumors, suggesting that optimal differentiation of tumors can indeed promote and maintain intact monocyte/osteoclast function. 

To understand the underlying mechanisms which govern inhibition of NK cell function by patient osteoclasts, we determined the surface expression of osteoclasts from cancer patients in comparison to healthy donors’ osteoclasts. The findings indicated that not only inhibitory MHC-class I expression is down-regulated, but also activating CD54, KLRG1, and MICA/B surface receptor expressions were decreased (Appendix A), which suggests an overall decrease in NK ligand expression. Loss of activating ligands could clearly be a reason for decreased activation of NK cells; however, loss of inhibitory receptors provides a more complex picture. Loss of expression of both activating and inhibitory NK cell ligands was also seen on osteoclasts from KC mice with pancreatic KRAS mutation correlating with the loss of NK cell function and generation of pancreatic tumors [36,37].

Supernatants from patient’s NK cells were less able to differentiate tumors, indicating that the function of secreted IFN-γ from patient NK cells is also severely compromised. Thus, pancreatic tumor induction and progression in patients is due to not only combined defects in NK expansion, decreased NK-cell mediated cytotoxicity and lower secretion of IFN-γ, and much lower ability of secreted IFN-γ to differentiate tumors, but also due to the defects in other subsets of immune cells which support NK cell expansion and function. 

Our studies indicate that immunotherapy by super-charged NK cells in the presence of AJ2 oral supplementation may not only restore immune function in cancer patients by delaying or curtailing the growth potential of poorly-differentiated/stem-like pancreatic tumors, but also by expanding and activating CD8+ T cells. This will not only allow NK-expanded CD8+ T cells to target NK-differentiated tumors, but, more importantly, they will add to the pool of differentiated tumors since NK-expanded CD8+ T cells can also produce IFN-γ and TNF-α upon activation. NK and CD8+ T cell-differentiated tumors can also be targeted by radiotherapeutic and/or chemotherapeutic strategies. 

## 4. Materials and Methods

### 4.1. Cell Lines, Reagents, and Antibodies

RPMI 1640 supplemented with 10% fetal bovine serum (FBS) (Gemini Bio-Products, San Diego, CA, USA) was used for the cultures of human NK cells. Human pancreatic cancer cell lines Panc-1, MIA PaCa-2 (MP2), BXPC3, HPAF, and Capan were generously provided by Dr. Guido Eibl (UCLA David Geffen School of Medicine) and PL12 was provided by Dr. Nicholas Cacalano (UCLA Jonsson Comprehensive Cancer Center). Panc-1, MP2, and BXPC3 were cultured with DMEM supplemented with 10% FBS and 1% Penicillin-Streptomycin (Gemini Bio-Products, West Sacramento, CA, USA). HPAF, Capan and PL12 were cultured in RMPI 1640 medium supplemented with 10% FBS and 1% penicillin-streptomycin. Recombinant human IL-2 was obtained from NIH-BRB. Human TNF-α and IFN-γ was obtained from Biolegend (San Diego, CA, USA). Antibody to CD16 was purchased from Biolegend (San Diego, CA, USA). Fluorochrome-conjugated human and mouse antibodies for flow cytometry were obtained from Biolegend (San Diego, CA, USA). Monoclonal antibodies to TNF-α and IFN-γ were prepared in our laboratory, and used at 1:100 dilutions to block rhTNF-α and rhIFN-γ functions. The human NK cell and monocyte purification kits were obtained from Stem Cell Technologies (Vancouver, BC, Canada). Propidium iodide (PI) and N-Acetyl Cysteine (NAC) were purchased from Sigma Aldrich (St. Louis, MO, USA). Cisplatin and paclitaxel were purchased from Ronald Reagan UCLA Medical Center Pharmacy (Los Angeles, CA, USA). 

### 4.2. Ethics Approval and Consent to Participate

Written informed consents approved by UCLA Institutional Review Board (IRB) were obtained and all procedures were approved by the UCLA-IRB (IRB#11-000781). Animal research was performed under the written approval of the UCLA Animal Research Committee (ARC) (protocol # 2012-101-13). 

### 4.3. Purification of Human NK Cells and Monocytes

NK cells and monocytes were negatively selected from PBMCs using isolation kits from Stem Cell Technologies (Vancouver, BC, Canada). Greater than 96% purity was obtained both for purified NK cells and monocytes based on flow cytometric analysis.

### 4.4. Analysis of Human Pancreatic Cancer Cell Growth in Immune-Deficient (NSG) and Humanized-BLT Mice

Humanized-BLT (hu-BLT; human bone marrow/liver/thymus) mice were generated as previously described [55,56]. In vivo growth of pancreatic tumors was performed by orthotopic tumor implantation in the pancreas of NSG or hu-BLT mice. To establish orthotopic tumors, mice were anesthetized using isoflurane, and tumors in a mixture with Matrigel (10 μL) (Corning, NY, USA) were injected in the pancreas using insulin syringe. Mice received 1.5 × 10^6^ super-charged NK cells via tail vein injection 7 to 10 days after the tumor implantation. They were also fed AJ2 (5 billion/dose) orally. The first dose of AJ2 was given one or two weeks before tumor implantation, and feeding was continued throughout the experiment at an interval of every 48 h. Mice were euthanized when signs of morbidity were evident. Pancreas, pancreatic tumors, bone marrow, spleen, and peripheral blood were harvested and single cell suspensions were prepared from each tissue as described previously [23] and below.

### 4.5. Cell Dissociation and Cell Culture of Tissues from hu-BLT and NSG Mice

Pancreatic tumors were harvested from NSG and hu-BLT mice and cut into 1 mm^3^ pieces and placed into a digestion buffer containing 1 mg/mL collagenase IV, 10 U/mL DNAse I, and 1% bovine serum albumin (BSA) in DMEM media for 20 min at 37 °C. The samples were then filtered through a 40 mm cell strainer and centrifuged at 1500 rpm for 10 min at 4 °C. To obtain single-cell suspensions from BM, femurs were flushed using media, and filtered through a 40 µm cell strainer. Spleens were removed and single cell suspensions were prepared and filtered through a 40 µm cell strainer and centrifuged at 1500 rpm for 5 min at 4 °C. The pellets were re-suspended in ACK buffer to remove the red blood cells. Peripheral blood mononuclear cells (PBMCs) were isolated using ficoll-hypaque centrifugation. 

### 4.6. Isolations of NK Cells, T Cells and Monocytes from hu-BLT Mice

NK cells and T cells from hu-BLT splenocytes were obtained as described previously by using the human CD56+ and CD3+ selection kits respectively (Stem Cells Technologies, Vancouver, BC, Canada). Monocytes from hu-BLT bone marrow were isolated using human CD14 isolation kit (eBioscience, San Diego, CA, USA) [23].

### 4.7. Generation of Osteoclasts and Expansion of Human and hu-BLT NK Cells

Monocytes were purified form human peripheral blood or hu-BLT BM and cultured using alpha-MEM medium containing M-CSF (25 ng/mL) and RANKL (25 ng/mL) for 21 days (medium was refreshed every 3 days). NK cells were activated with rh-IL-2 (1000 U/mL) and anti-CD16 mAb (3 μg/mL) for 18–20 h before they were cultured with osteoclasts and sonicated-AJ2 to generate super-charged NK cells. The medium was refreshed every 3 days with RMPI containing rh-IL-2 (1000 U/mL).

### 4.8. In-Vitro MP2 and OSCSCs Cancer Stem Cell Differentiation

Differentiation of MP2 and OSCSCs (oral squamous carcinoma stem-cells) tumors was conducted as described previously [22,49]. Briefly NK cells were treated with a combination of anti-CD16 mAb (3 μg/mL) and IL-2 (1000 U/mL) for 18 h before the supernatants were removed and used for differentiation of the tumors. The amounts of IFN-γ produced by activated NK cells were assessed using ELISA kits purchased from Biolegend (San Diego, CA, USA). To induce differentiation of tumors a total of 3500 pg of IFN-γ containing supernatants were added for 4 days.

### 4.9. Enzyme-Linked Immunosorbent Assays (ELISAs) and Multiplex Cytokine Assay

Human ELISA kits for IFN-γ and IL-6 were purchased from Biolegend (San Diego, CA, USA). The assays were conducted as recommended by the manufacturer. For certain experiments multiplex arrays were used to determine the levels of secreted cytokines and chemokines. Analysis was performed using MAGPIX (Millipore, Danvers, MA, USA) and data was analyzed using xPONENT 4.2 (Luminex, Austin, TX, USA).

### 4.10. Surface Staining and Cell Death Assays

Staining was performed by staining the cells with antibodies as described previously [57,58,59], briefly, antibodies were added to 1 × 10^4^ cells in 50 µL of cold-PBS+ 1% BSA and cells were incubated on ice for 30 min. Thereafter cells were washed in cold PBS+ 1% BSA and flow cytometric analysis was performed using Beckman Coulter Epics XL cytometer (Brea, CA, USA) and results were analyzed in FlowJo vX software (Flowjo, Ashland, OR, USA).

### 4.11. ^51^Cr Release Cytotoxicity Assay

The ^51^Cr release assay was performed as described previously [60]. Patient-derived OSCSCs were used as a specific and sensitive NK targets to assess NK cell-mediated cytotoxicity [49]. Briefly, different numbers of effector cells were incubated with ^51^Cr–labeled OSCSCs. After 4 h incubation the supernatants were harvested from each sample and counted on a gamma counter. The percentage specific cytotoxicity was calculated using the following formula:(1)$$\%\;\mathrm{Cytotoxicity}=\frac{\mathrm{Experimental}\;\mathrm{cpm}-\mathrm{Spontaneous}\;\mathrm{cpm}}{\mathrm{Total}\;\mathrm{cpm}-\mathrm{Spontaneous}\;\mathrm{cpm}}$$

Lytic unit 30/10^6^ is calculated by using the inverse of the number of effector cells needed to lyse 30% of tumor target cells × 100.

### 4.12. Statistical Analysis

An unpaired, two-tailed Student t-test was performed for the statistical analysis. One-way ANOVA using Prism-7 software (Graphpad Prism, San Diego, CA, USA) was used to compare different groups. (*n*) denotes the number of mice used for each condition in the experiment. The following symbols represent the levels of statistical significance within each analysis, *** (*p*-value < 0.001), ** (*p*-value 0.001–0.01), * (*p*-value 0.01–0.05).

## 5. Conclusions

Although the role of NK cells in targeting metastatic tumors has been known for a long time, the mechanisms underlying the clearance of such tumors have not been clearly delineated. Most previous reports have focused on the killing ability of NK cells, and at times they have singled out this important function of NK cells as perhaps the sole reason for the elimination of tumors. However, the study reported in this paper indicates that both lysis and differentiation of tumors by the NK cells are important mechanisms by which NK cells are capable of preventing the induction and progression of tumors. 

Use of murine models for the studies of NK cells has paved the road for significant discoveries, and these models were instrumental in driving progress in the field. However, several differences have been identified between the human and murine NK function and phenotype which cast a shadow over the application of the findings from the murine model to human diseases [61]. Thus, we need to find strategies that appropriately model human diseases. The generation of humanized mice models is a step forward towards such a goal. Although, some differences have previously been identified between the humanized mouse models and humans, overall they serve as appropriate models to study human cancers, since we have found good agreement between ex vivo data generated from human cancer patients and those found from humanized mice models implanted with the human tumors. Moreover, many outcomes which are observed during the interaction of NK cells with tumors in vitro, are also observed in vivo in humanized mice implanted with stem like/undifferentiated tumors.

Our studies indicated that an intact immune system is required for the elimination of tumors. However, tumors have been shown to cause immune suppression, in particular NK suppression, and this defect occurs in NK cells at many levels. NK cells from both cancer patients and humanized mice implanted with tumor lose their ability to kill and differentiate tumors. The inability of NK cells to curtail tumor growth through increased lysis and differentiation of tumors is a profound deficiency which will require significant intervention. Such intervention could be through the administration of super-charged NK cells, as we have seen in hu-BLT mice implanted with poorly differentiated pancreatic tumors. Admittedly, the human body is much larger and perhaps more complex and findings from the tumor bearing humanized mice will need to be tested in humans. Nevertheless, the findings from our study will pave the road for the future treatment strategies in humans.

## Figures and Tables

**Figure 1 cancers-12-00063-f001:**
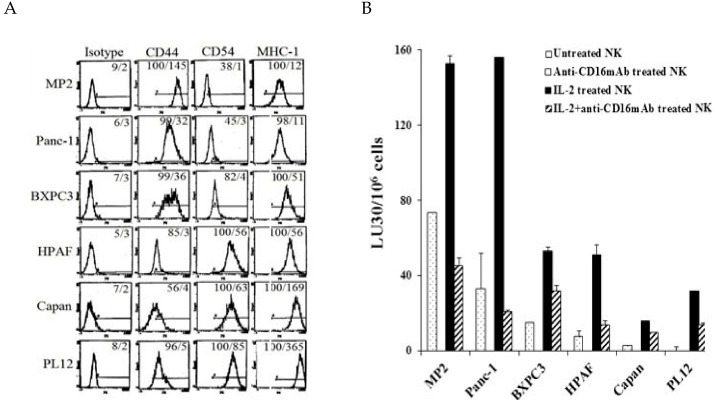
The stage of differentiation in pancreatic tumors correlated with susceptibility to NK cell-mediated cytotoxicity. The surface receptor expression of CD44, CD54, and MHC-class I on pancreatic tumors were assessed by flow cytometry after staining with PE-conjugated antibodies. Isotype control antibodies were used to determine non-specific binding. Numbers in each histogram represent percent/mean channel fluorescence (MFI). (**A**) Freshly isolated NK cells were left untreated or treated with anti-CD16 mAb (3 μg/mL), IL-2 (1000 U/mL), or the combination of anti-CD16 mAb (3 μg/mL) and IL-2 (1000 U/mL) for 18 h before they were added to the cultures of ^51^Cr labeled MP2, Panc-1, BXPC3, HPAF, Capan, or PL12. NK cell-mediated cytotoxicity was determined using 4 h ^51^Cr release assay, and the lytic units 30/10^6^ cells were determined using inverse numbers of NK cells required to lyse 30% of the target cells × 100. (**B**) One of several representative experiments is shown in the figure (please also see Appendix A).

**Figure 2 cancers-12-00063-f002:**
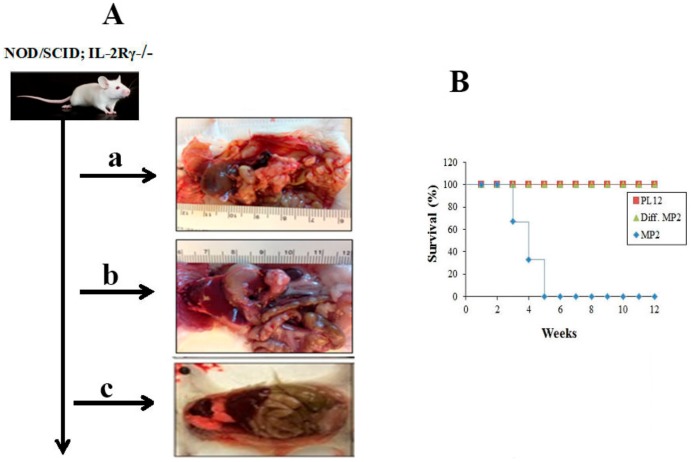
Lack of tumor growth, and long-term survival of NSG mice after orthotopic implantation of NK-differentiated MP2 tumors in pancreas; MP2 tumors were differentiated by the NK-supernatants as described in the Materials and Method section. MP2 tumors (3 × 10^5^) (*n* = 3) (panel **a**), patient-derived differentiated PL12 (2 × 10^6^) (*n* = 3) (panel **b**), and NK-differentiated MP2 tumors (diff-MP2) (5 × 10^5^) (*n* = 3) (panel **c**), were implanted into the pancreas of NSG mice and tumor growth were determined in 4 weeks for MP2 tumors and 12 weeks for PL-12 and diff-MP2 tumors (**A**). The rates of survival of the mice in panels a, b and c (**B**) as well as tumor metastasis to liver (Appendix A) were determined after euthanasia.

**Figure 3 cancers-12-00063-f003:**
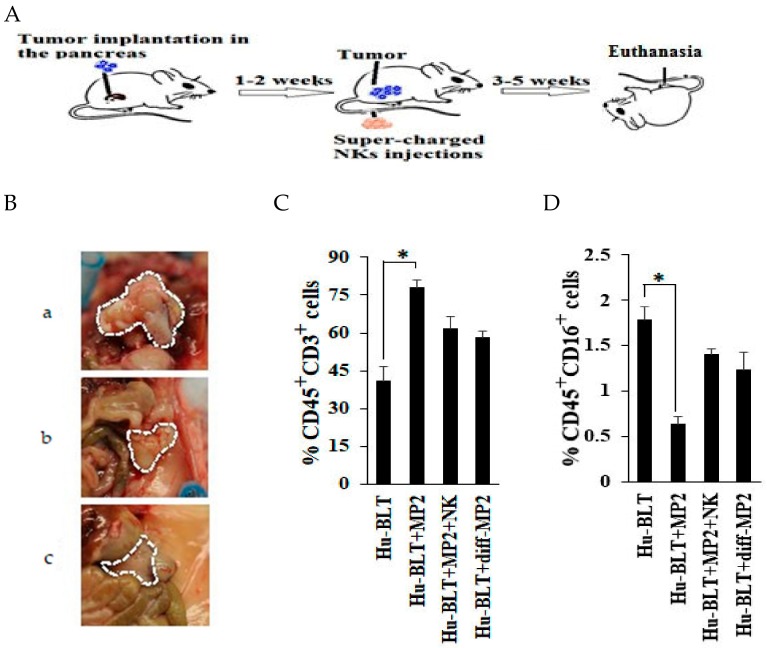
Single injection of super-charged NK cells inhibited tumor growth and increased immune cells in the pancreas in hu-BLT mice. Hu-BLT mice were generated as described in the Materials and Methods section and as depicted in the Appendix A in Appendix A, and they were implanted with 1 × 10^6^ tumors in the pancreas, and, injected with 1.5 × 10^6^ super-charged NK cells via tail vein after one to two weeks (**A**) and disease progression was monitored. After 6–7 weeks mice were euthanized and pictures of the tumors within the pancreas were taken post-mortem in mice injected with tumor in the absence of super-charged NK cells (panel **a**), tumor-bearing mice injected with super-charged NK cells (panel **b**), and NK-differentiated tumors (panel **c**) (*n* = 5/each experimental condition); one representative experiment is shown in the figure (**B**). Pancreas was dissociated as described in the materials and methods section and the single cells (1 × 10^6^ cells/ml) were used to determine the percentages of huCD45+CD3+ (**C**) and huCD45+CD16+ cells (**D**) (*n* = 3/each experimental condition).

**Figure 4 cancers-12-00063-f004:**
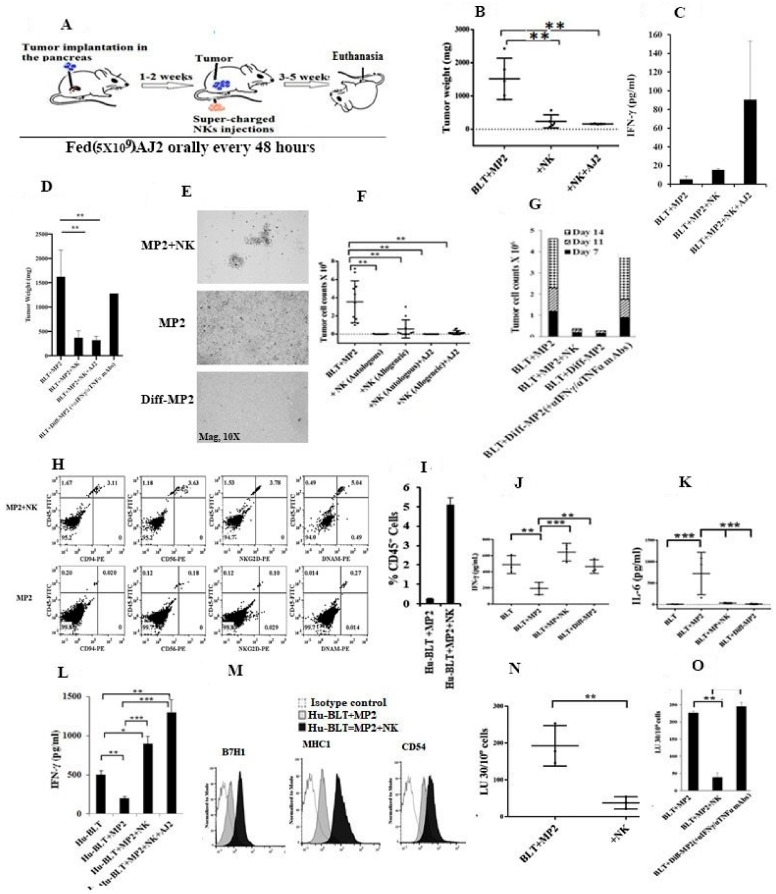
Single injection of super-charged NK-cells with/without feeding with AJ2 inhibited tumor growth due to differentiation of tumors in hu-BLT mice. Implantation of tumor cells in the pancreas and tail vein injection of super-charged NK cells and feeding with AJ2 (5 × 10^9^) were carried out as depicted in the figure (**A**), and disease progression was monitored. Mice were sacrificed, and pancreatic tumors were harvested and weighed (*n* = 4/each experimental condition) (**B**). Implantation of tumor cells in the pancreas and tail vein injection of super-charged NK cells were carried out as depicted in Figure 4A, and at the time of sacrifice mice were bled and the levels of IFN-γ in the serum were determined using multiplex array (*n* = 3) (**C**). Procedures were carried out as depicted in Figure 4A, Appendix A and Appendix A before pancreatic tumors were harvested and weighed (*n* = 3) (**D**). Upon sacrifice, pancreatic tumors were harvested and single cell suspensions were prepared as described in Materials and Methods. The same numbers of pancreatic tumor cells from each mouse were cultured, and the pictures of cultured tumors were taken on day 7. One of the four representative experiments is shown in the figure (**E**). Procedures were carried out as described in Figure 4A using injections of allogeneic or autologous super-charged NK cells. Pancreatic tumors were resected and single-cell suspensions were prepared and tumor growth were assessed (*n* = 9 to 12/each experimental condition) using identical numbers of tumors cultured from each mouse tumor (**F**). Hu-BLT mice were implanted with MP2 tumors and injected with NK cells or implanted with NK-differentiated tumors as described in Figure 4A, and Appendix A. At the end of the experiment pancreatic tumors were harvested and tumor growth was assessed on days 7, 11 and 14, and on day 7 attached tumors from each well were counted and equal numbers of tumors from each group were re-cultured and tumor growth in each well was determined every 3 days (*n* = 12/each experimental condition, one representative experiment is shown in the figure) (**G**). Hu-BLT mice were implanted with tumors and injected with super-charged NK cells, as described in Figure 4A. Tumors were resected, and single cell cultures were prepared and cultured for 7 days, after which percentages of human CD45, CD94, CD56, NKG2D, and DNAM within the tumors were determined after staining with antibodies, followed by flow cytometric analysis (**H**,**I**, Appendix A). Procedures were carried out as described in Figure 4A and Appendix A. Pancreata were removed, and single cell cultures were prepared and equal numbers of pancreatic cells were cultured until day 7 or 11 and the levels of IFN-γ (**J**) and IL-6 (**K**) were determined in culture supernatants (*n* = 4/each experimental condition). Hu-BLT mice were implanted with MP2 tumors and injected with NK cells and fed with AJ2 as described in Figure 4A. Upon sacrifice, tumors were resected, and single cell cultures were prepared, and equal numbers of tumors were cultured on day 7 or 11 and the levels of IFN-γ were determined in culture supernatants (*n* = 2/each experimental condition) (**L**). Procedures were carried out as described in Figure 4A. Tumors were resected, and single cell cultures were prepared and surface expressions of MHC-class I, B7H1 and CD54 were determined on tumors after culture (*n* = 4/each experimental condition) (**M**). NK cells (1 × 10^6^ cells/mL) from healthy individuals were treated with IL-2 (1000 U/mL) for 18–24 h before they were added to ^51^Cr labeled tumors obtained from mice implanted with different tumors and/or injected with NK cells as described in Figure 4A at various effector to target ratios. NK cell mediated cytotoxicity was determined using 4 h ^51^Cr release assay, and the lytic units 30/10^6^ cells were determined using inverse number of NK cells required to lyse 30% of the tumor-cells × 100 (*n* = 3/each experimental condition) (**N**,**O**). The following symbols represent the levels of statistical significance within each analysis, *** (*p*-value < 0.001), ** (*p*-value 0.001–0.01), * (*p*-value 0.01–0.05)

**Figure 5 cancers-12-00063-f005:**
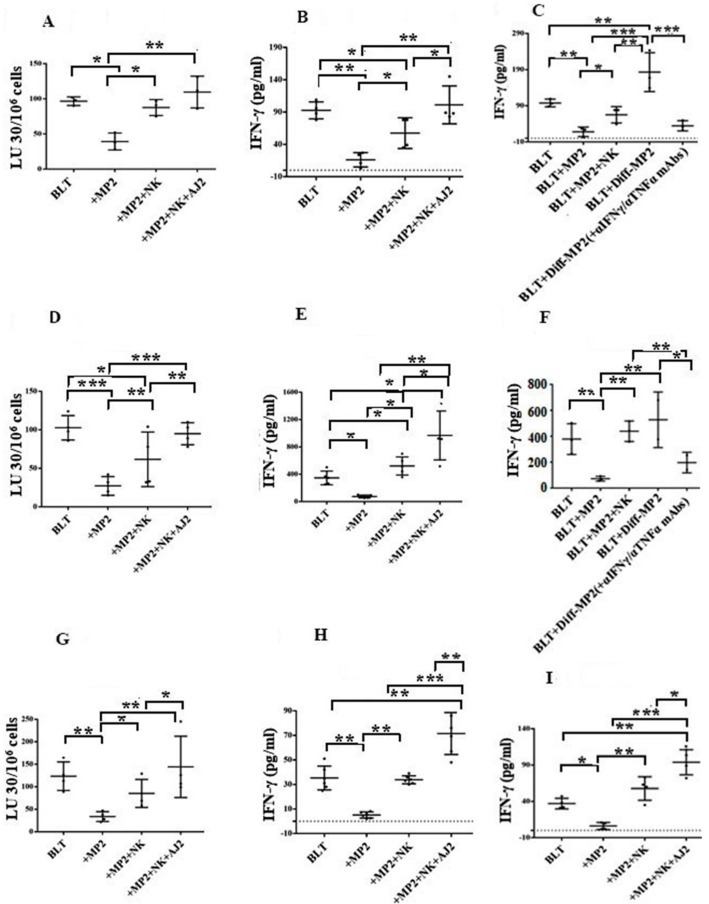
Injection of super-charged NK-cells with/without feeding with AJ2 restored and increased IFN-γ secretion and/or cytotoxic function of NK cells from different tissues of tumor-bearing hu-BLT mice. Procedures were carried out as described in Figure 4A, Appendix A. Upon sacrifice, PBMCs were isolated from blood and treated with IL-2 (1000 U/mL) before they were used in cytotoxicity assay against OSCSCs using 4 h ^51^Cr release assay. Lytic units 30/10^6^ cells were determined using inverse number of NK cells required to lyse 30% of the target cells × 100 (**A**). Procedures were carried out as described in Figure 4A, and Appendix A before the PBMCs were isolated and treated with (1000 U/mL) and the supernatants were harvested and IFN-γ secretion was determined using ELISA (*n* = 4/each experimental condition) (**B**,**C**). Procedures were carried out as described in Figure 4A, and Appendix A before spleens were harvested, and single-cell suspensions were prepared. Splenocytes were treated with IL-2 (1000 U/mL) before they were used for cytotoxicity against OSCSCs using 4 h ^51^Cr release assay. Lytic units 30/10^6^ cells were determined using inverse number of NK cells required to lyse 30% of the target cells × 100 (*n* = 4/each experimental condition) (**D**). Procedures were carried out as described in Figure 4A, and Appendix A before the supernatants were harvested from day 3 or 7 cultures of splenocytes, and IFN-γ secretion was determined using ELISA (*n* = 5/each experimental condition) (**E**,**F**). Procedures were carried out as described in Figure 4A and NK-enriched cells were isolated from splenocytes and were cultured with IL-2 (1000 U/mL) before they were used for cytotoxicity against OSCSCs using 4 h ^51^Cr release assay. Lytic units 30/10^6^ cells were determined using inverse number of NK cells required to lyse 30% of the target cells × 100 (*n* = 4/each experimental condition) (**G**). Supernatants were harvested from day 3 or 7 NK-enriched cultures and IFN-γ secretion was determined using ELISA (*n* = 6/each experimental condition) (**H**). Procedures were carried out as described in Figure 4A and the CD3+ T-cells were isolated from splenocytes and were cultured with IL-2 (100 U/mL), and on day 3 or day 7 the supernatants were harvested and IFN-γ secretion was determined using ELISA (*n* = 4/each experimental condition) (**I**). Procedures were carried out as described in Figure 4A, and Appendix A and BM cells were harvested and treated with IL-2 (1000 U/mL) for 7 days before they were used for cytotoxicity against OSCSCs using 4 h ^51^Cr release assay. Lytic units 30/10^6^ cells were determined using inverse number of NK cells required to lyse 30% of the target cells × 100 (*n* = 4/each experimental condition) (**J**). Supernatants were harvested on day 3 or day 7 of BM cultures and IFN-γ secretion was determined using ELISA (*n* = 6/each experimental condition) (**K**,**L**). The following symbols represent the levels of statistical significance within each analysis, *** (*p*-value < 0.001), ** (*p*-value 0.001–0.01), * (*p*-value 0.01–0.05)

**Figure 6 cancers-12-00063-f006:**
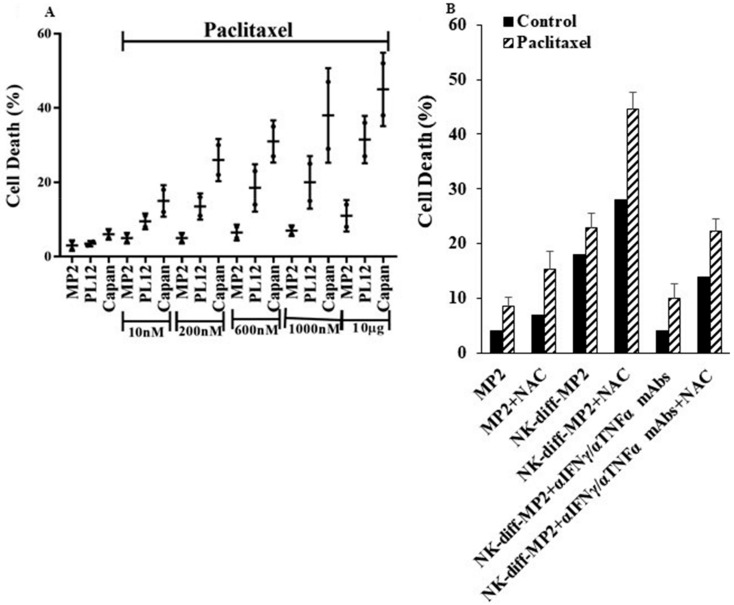
Paclitaxel induced significant cell death in patient-derived differentiated PL12 and Capan and NK-differentiated MP2 tumors treated with and without NAC. MP2, PL12, and Capan tumors (1 × 10^5^ tumors) were treated with or without Paclitaxel for 18–20 h before the viability of cells was determined using propidium iodide staining. *p*-values of <0.05 were obtained for differences between MP2 vs. PL12 and Capan at the concentrations of 10, 200, 600, and 1000 nM and 10 g of Paclitaxel (*n* = 2/each experimental condition) (**A**). MP2 tumors were differentiated with NK supernatants in the presence and absence of anti-IFN-γ and anti-TNF-α as described in Materials and Methods section for a period of 5 days before they were washed and treated with/without NAC (20 nM) for 24 h, followed by paclitaxel treatment from 10 nM to 600 nM for 18–24 h. The viability of cells was determined by staining with propidium iodide (*n* = 3/each experimental condition) (**B**).

**Figure 7 cancers-12-00063-f007:**
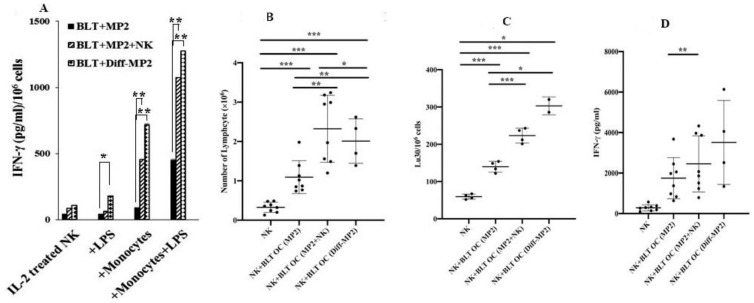
Monocytes or osteoclasts from tumor-bearing mice injected with NK cells or implanted with NK-differentiated MP2 tumors induced increased IFN-γ secretion by the NK cells when compared to those of tumor-alone implanted mice. Hu-BLT mice were implanted with tumors and injected with NK cells, as described in Figure 4A before spleen and BM were harvested and single cell suspensions were prepared. CD56+NK cells were positively selected from splenocytes, and monocytes were purified from the BM cells, and co-cultured at (NK:Monocytes; 2:1 ratio) and treated with IL-2 (1000 U/mL) alone or in combination with LPS (100 ng/mL) for 7 days before the supernatants were harvested and IFN-γ secretion was determined using ELISA. One of three representative experiments is shown (**A**). Osteoclasts were generated from monocytes isolated from the BM of hu-BLT mice. Allogeneic NK cells purified from healthy individuals were treated with IL-2 (1000 U/mL) and anti-CD16 mAb (3 µg/mL) for 18 h before they were either cultured alone or in the presence of hu-BLT-OCs and sAJ2 (NK:OCs:sAJ2; 2:1:4), and the numbers of expanding NK cells were determined on days 6, 9, 12, and 15. At each day of culture, equal numbers of NK cells from each group were cultured and their cell growth was determined (*n* = 4 to 8) (**B**). On day 15 of the culture, cells were counted, and equal numbers of NK cells were used for cytotoxicity against OSCSCs using 4-h ^51^Cr release assay. Lytic units 30/10^6^ cells were determined using inverse number of NK cells required to lyse 30% of the target cells × 100 (*n* = 4 to 8) (**C**). The supernatants from the NK and OC cultures as described in Figure 7B were harvested on days 6, 9, 12, and 15, and the levels of IFN-γ secretion were determined using ELISA (*n* = 4 to 8) (**D**). The following symbols represent the levels of statistical significance within each analysis, *** (*p*-value < 0.001), ** (*p*-value 0.001–0.01), * (*p*-value 0.01–0.05)

## Data Availability

Data generated or analyzed during the study are included in this submitted article (main manuscript and Appendix A).

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
