# Peer review of "Probiotic-Treated Super-Charged NK Cells Efficiently Clear Poorly Differentiated Pancreatic Tumors in Hu-BLT Mice"

_cancers, 2019, doi:10.3390/cancers12010063_

Round 1

Reviewer 1 Report

The authors employ a strategy to expand supercharged NK cells by using osteoclasts as feeder cells in the presence of sAJ2.  They show that NK cells aid in the differentiation of pancreatic cancer stem cells which decreases these cells growth, metastatic potential, and ultimately lead to a better response to chemotherapy and disease prognosis.  This was shown in vitro and in vivo.

Major comments:

For Figures 1B, 4C, 4D, 7A, 7B, 7C, and 7D, I’m a little concerned that “one ....representative experiment is shown”.  Can you really show statistical significance in these because you are showing biological replicates and not true experimental replicates.  If the data are all statistically significant then you should be able to pool all of the data and get means +/- SEMs that are relevant (even though in vivo increases the variability).  Please show data that are true for the overall trend and not one of numerous experiments.  It makes me worry that the authors are hiding insignificant data and only showing one experiment that supports their claims.

Please expand on the discussion.  The authors do a great job summarizing the extensive data but do not provide much support with how this fits with the current literature nor pose any “big picture” mechanism(s).   For example, do the NK cells alter the epigenome of the CSCs?  The results summarize the data and discussions are meant to discuss and theorize what the data mean in the larger sense and how they fit in the known literature.

Minor comments:  

“has” is misspelled in the introduction at “The use of allogenic NK cells....”

Please define the abbreviations: GVHD and NAC in the text or in the abbreviations list.

Author Response

For Figures 1B, 4C, 4D, 7A, 7B, 7C, and 7D, I’m a little concerned that “one ....representative experiment is shown”.  Can you really show statistical significance in these because you are showing biological replicates and not true experimental replicates.  If the data are all statistically significant then you should be able to pool all of the data and get means +/- SEMs that are relevant (even though in vivo increases the variability).  Please show data that are true for the overall trend and not one of numerous experiments.  It makes me worry that the authors are hiding insignificant data and only showing one experiment that supports their claims.

We thank the reviewer for the suggestion, and we have now changed those figures and added more experiments, and shown the statistical analysis. As the reviewer correctly pointed out this increases the variability. In addition, functional assays such as 51Cr release assay and ELISAs because of variability at many different levels, such as the levels of chromation of the cells, sensitivity of the targets, etc it does make it difficult to compile different experiments run on different days due to increase variability. However, for experiment Fig. 1B we have now included 5 independent experiments for the MP2 and PL12, the two tumor types that we used in our experiments and performed statistical analysis. The results are shown in supplemental since this is a confirmatory result. We have also added new figures with statistical analysis for the other figures requested above.

Please expand on the discussion.  The authors do a great job summarizing the extensive data but do not provide much support with how this fits with the current literature nor pose any “big picture” mechanism(s).   For example, do the NK cells alter the epigenome of the CSCs?  The results summarize the data and discussions are meant to discuss and theorize what the data mean in the larger sense and how they fit in the known literature.

As suggested, we have expanded on the discussion and included more discussion points on the big picture in the discussion and in the overall conclusion of paper. Since these are novel studies on using hu-BLT mice model to study the concept of NK cells as important effectors in differentiation of the tumors, we do not have publications from other laboratories to cite and discuss, even though the concept is well established in our laboratory due to large numbers of manuscripts we published using in vitro assays previously (we have cited many in this manuscript). In addition, we do not yet have evidence regarding the role of NK cells in modification of epigenome in our model. We are currently working on delineating the signaling requirements for the differentiation and since this manuscript is large as is, we will report these important studies in a subsequent report. We hope we have addressed the reviewer’s concern satisfactorily.

Minor comments:  

“has” is misspelled in the introduction at “The use of allogenic NK cells....”

We have corrected it. Thanks

Please define the abbreviations: GVHD and NAC in the text or in the abbreviations list.

We have defined the abbreviations in the abbreviation list

Reviewer 2 Report

Dear Sirs,

The manuscript entitled “Super-charged NK cells for the treatment of pancreatic cancer; role in lysis and differentiation of poorly differentiated tumors in pancreas of hu-BLT mice by Kawaljit Kaur, and collaborators,  essentially describes the implantation and characterization of  a humanized mice as a platform to investigate the role of NK cells in treatment of pancreatic differentiated and undifferentiated tumors. Studies carried out using mice have provided valuable insights into the tumor and immune cell interactions. Therefore the subject of the manuscript is pertinent and it seems to be experimentally correct, therefore deserving publication after a few improvements.

Comments:

- In Results authors refer that “Six different pancreatic tumor types, each characterized at poorly, intermediate and well differentiated stages pathologically by other laboratories previously [54] were used to determine phenotype”. I think that a better characterization of the differences between the used undifferentiated and differentiated tumors, beyond that was referred (i.e levels of MHC class I and CD54 in the presence of higher surface expression of CD44), would improve the manuscript. In fact, the available published data indicates that NK cells interact dynamically with the tumor microenvironment as for example, cellular components of stroma, extracellular matrix and local immune cells and are highly sensitive to this crosstalk.

-Sometimes the description of the results is arid and some discussion including data available in literature for similar approaches could improve it, highlighting the real contribution of the present work without overloading the main discussion.

Author Response

- In Results authors refer that “Six different pancreatic tumor types, each characterized at poorly, intermediate and well differentiated stages pathologically by other laboratories previously [54] were used to determine phenotype”. I think that a better characterization of the differences between the used undifferentiated and differentiated tumors, beyond that was referred (i.e levels of MHC class I and CD54 in the presence of higher surface expression of CD44), would improve the manuscript. In fact, the available published data indicates that NK cells interact dynamically with the tumor microenvironment as for example, cellular components of stroma, extracellular matrix and local immune cells and are highly sensitive to this crosstalk.

We thank the reviewer for his/her suggestions. As for further characterization, in our previous publications we have done in depth phenotypic and functional analysis of MP2 and PL-12, the two tumor types that we use in this manuscript as representative for pancreatic tumors ([1-6]                               . In addition, this concept does not only apply to pancreatic tumors but also we have seen in Oral [1-6] Glioblastoma [7], and more recently in Lung tumors, and in all cases we see similar profile for the tumors, and identical functional outcome for NK cells. Interestingly, we see similar profiles when we knock down a gene in Breast and Melanoma tumors in which case we convert more differentiated tumors to a less differentiated tumor and this increases the NK activity significantly [1, 6]. Therefore, this concept is not limited to pancreatic tumors but it is a common feature among many different tumor types.

As to microenviornmental influence on NK cells the reviewer is absolutely correct since the microenvironment has a significant effect on NK cells and this effect is important in shaping the phenotype and function of NK cells. I have recently written many reviews regarding this point  [8-10], and we have also shown previously that healthy stem cells such as Dental Pulp Stem Cells, iPSCs, Embryonic Stem cells and Mesenchymal Stem cells all have similar phenotype to those of tumors in terms of surface receptor expression and effect on NK cells. In addition, many other immune effectors such as monocytes, osteoclasts and Dendritic cells as shown in our previous publications and in this report to have profound effect on NK cells, and indeed we have written many reviews in this regard. Unfortunately, we have certain page limits which we need to abide by and therefore, we could not include many important points in the discussion, however, as suggested by the reviewer we have included a paragraph in this regard in the discussion section.

-Sometimes the description of the results is arid and some discussion including data available in literature for similar approaches could improve it, highlighting the real contribution of the present work without overloading the main discussion.

Although our approach is very novel in the field, we have identified several publications which have shown agreement with our findings and included those in the discussion section. We have not found any publications which argue against our concept.

Reviewer 3 Report

Cancers, MS # 657728. This manuscript describes the efficacy of pro-biota-treated NK cells against undifferentiated or differentiated pancreatic tumor cells using in vivo hu-BLT mice model. Authors present extensive data with appropriate approach and explanations. This manuscript requires moderate level of modifications before further considerations.

Title is confusing. Please consider changing the title. Suggestion, “Pro-biota-treated human NK cells efficiently clear undifferentiated pancreatic tumor in hu-BLT mice” Abstract: A clear definition should be given for the ‘NK-differentiated’ phrase. Abstract needs to be tightened. The immunological relevance of ‘undifferentiated’ and differentiated’ pancreatic tumors should be stated as part of the function of the NK cells. It is not clear why it matters. Results section of the abstract should be improved and simplified. Introduction can be reorganized to better explain: 1) undifferentiated and differentiated pancreatic tumors and 2) Role of NK cells in both helping to differentiate the tumor cells and their role in clearing the tumor cells. i) Page-2, Para-3, Line #8. ‘has’ and not ‘hase’. ii) Other examples are ‘anti-CD16mAb’-need space between CD16 and mAb; ‘mAb treated’ should be hyphenated. iii) Isotype ‘control’ antibodies were used as ‘controls’. Authors should go through the text and carefully check the text. Was there a quantification for the flow data (MFI) presented in Figure-1? What are the numbers presented with in each flow histograms? No description of this given within the figure legend. Following treatment of NK cells (Figure 1B), was there a difference in the NK cell viability? Figure 2. Left panels that shows all the organs in this figure do not contribute to the overall understanding. Bottom middle panel appears to be a spleen and not pancreas. Authors need to clarify. Right panels do not evenly portray the livers. Did the authors perform histological evaluations of these organs (trichrome staining)? Panels should be individually labeled (A, B..). Figure 3. Panel B should be trimmed and only the pancreas should be shown. Individual panels should be labeled only A, B, C etc. Have the authors characterized the contents of the NK cell-derived supernatant? Do the authors know other soluble factors apart from the presence of IFN-g and TNF-a? Figure 3. Did the authors further dissect the CD3+ T cells? Such as CD4, CD8, Th17 etc? Authors define an increase in CD45+ ‘immune cells’. What do they mean by this? CD45 defines all the leukocytes (or all the hematopoietic cells except erythrocytes). Such a statement is confusing. Text describing the CD45+ cells and the data in the figure (CD45+CD3+ or CD45+CD16+) are not matching and confusing. Authors need to fix this. Figure 3. Did the authors quantify the size or the weight of the tumors in Figure 3? Figure 4. Change the labels to A, B, C etc. Include the number of AJ2 orally ingested in panel-A. Are there error bars in panel-C? If four mice per condition was used, did the authors calculate the SEM? Figure 4. Did the authors calculate the significance and the SEM in panel-D for the tumor weight? Figure 4E. These microscope images are difficult to see and interpret. Authors should provide better images with incorporated scale bars. Do the pattern of cell clustering in these images represent the stem-like (undifferentiated) and differentiated states of the tumor cells? Can the authors elaborate on this, as the top image shows ‘cluster or colony’ and not the middle or the bottom image? Figure 4H. Numbers within the panels and labels outside the panels are difficult to read. Figure H and I are not convincing. Figure 5. Change panel labels. Do panels J and K require labels ‘BM’? References #44 and #48 are duplicates. Please check all the references.

Author Response

Title is confusing. Please consider changing the title. Suggestion, “Pro-biota-treated human NK cells efficiently clear undifferentiated pancreatic tumor in hu-BLT mice”

We have modified the title as suggested. Thank you

Abstract: A clear definition should be given for the ‘NK-differentiated’ phrase. Abstract needs to be tightened. The immunological relevance of ‘undifferentiated’ and differentiated’ pancreatic tumors should be stated as part of the function of the NK cells. It is not clear why it matters.

We have defined “NK-differentiated” and added a couple of sentences to demonstrate the relevance of undifferentiated pancreatic tumors to NK function. We hope we have addressed the concern of the reviewer satisfactorily.

Results section of the abstract should be improved and simplified.

We have modified the result section to improve and simplify

Introduction can be reorganized to better explain: 1) undifferentiated and differentiated pancreatic tumors and 2) Role of NK cells in both helping to differentiate the tumor cells and their role in clearing the tumor cells.

As suggested, we reorganized the introduction to explain 1) undifferentiated and differentiated pancreatic tumors and 2) Role of NK cells in both helping to differentiate the tumor cells and their role in clearing the tumor cells.

Page-2, Para-3, Line #8. ‘has’ and not ‘hase’.

We have corrected it.

Other examples are ‘anti-CD16mAb’-need space between CD16 and mAb; ‘mAb treated’ should be hyphenated.

We have corrected as suggested

Isotype ‘control’ antibodies were used as ‘controls’. Authors should go through the text and carefully check the text. Was there a quantification for the flow data (MFI) presented in Figure-1? What are the numbers presented with in each flow histograms? No description of this given within the figure legend.

We have corrected as suggested, and added description for the quantification of flow data. Numbers in each histogram represent the percent/Mean Channel Fluorescence (MFI).

Following treatment of NK cells (Figure 1B), was there a difference in the NK cell viability?

The only treatment that affected viability of NK cells were treatment with IL-2+anti-CD16 mAb in which we observed an increase in cell death as reported also in our previous publications. However, for each experiment we always counted the viable cells and adjusted the numbers based on viable NK cells before their culture with tumor cells, therefore, the effect is related per viable cell basis  

Figure 2. Left panels that shows all the organs in this figure do not contribute to the overall understanding.

We have modified the figure and taken out the organs and now we are showing them in the supplemental since we discuss about metastasis and we can clearly see metastasis with the MP2 and not with PL12 or NK differentiated tumor. However, as suggested we have taken the organs out from the figure.

Bottom middle panel appears to be a spleen and not pancreas.

It is a spleen on top of the pancreas, and both are healthy. However, we have now taken this picture and transferred to the supplemental.

Authors need to clarify. Right panels do not evenly portray the livers. Did the authors perform histological evaluations of these organs (trichrome staining)?

We apologize for the lack of even portrayal of the livers. The pictures were taken under different lighting conditions and different distances. We did do histological evaluation for certain conditions, however, since the metastasis was so prominent in the MP2 implanted (white patches), and healthy livers were observed with PL-12 and NK-differentiated tumors visually, we did not include them. In addition, to avoid too many figures we limited the presented panels to most important panels.

Panels should be individually labeled (A, B..).

As suggested, we have now labeled the panels individually A, B and C in the main manuscript.

Figure 3. Panel B should be trimmed and only the pancreas should be shown.

As suggested, we have trimmed and kept the scales of trimming similar to demonstrate the differences in the size of the pancreas since it reflects also the size of the tumors.

Individual panels should be labeled only A, B, C etc.

We have labeled individual panels A, B and C

Have the authors characterized the contents of the NK cell-derived supernatant?

Yes, we have in several previous publications. Indeed, we are continually studying the differences between primary NK cells and super-charged NK cells, and in recent experiments we have also conducted proteomics and single cell genomic and proteomic assays and found significant differences between the two and we will be reporting those results in our future papers.

Do the authors know other soluble factors apart from the presence of IFN-g and TNF-a?

Yes, many factors are increased in super-charged NK cells, however, we have also studied the role of increased IL-6 and IL-8 in NK cells and did not find any relevance to differentiation of tumors. We have reported those findings in a previous paper [7].

Figure 3. Did the authors further dissect the CD3+ T cells? Such as CD4, CD8, Th17 etc?

Yes, we have done extensive analysis of CD3+ T cells and found that NK cells increase the percentages of CD8+ T cells and limit CD4+ T cells. We are in the process of writing that manuscript and will be submitting soon. We have not determined Th17 cells.

 Authors define an increase in CD45+ ‘immune cells’. What do they mean by this? CD45 defines all the leukocytes (or all the hematopoietic cells except erythrocytes). Such a statement is confusing.

We intended to mention that analysis was performed by using CD45+ cell surface receptor, and that represents leukocytes. We have changed that in the text to CD45+ cells.

Text describing the CD45+ cells and the data in the figure (CD45+CD3+ or CD45+CD16+) are not matching and confusing. Authors need to fix this.

We apologize for this and we have completely changed the text and described it to match with the figure. We hope we have addressed it satisfactorily.

Did the authors quantify the size or the weight of the tumors in Figure 3?

Yes, they are included in the panels shown in Figure 4.

Figure 4. Change the labels to A, B, C etc. Include the number of AJ2 orally ingested in panel-A.

We have included the number of AJ2 in panel A

 Are there error bars in panel-C? If four mice per condition was used, did the authors calculate the SEM?

Yes, we have included the error bars in the panel.

Figure 4. Did the authors calculate the significance and the SEM in panel-D for the tumor weight?

Yes, we have now included the error bars and significance in panel D

Figure 4E. These microscope images are difficult to see and interpret. Authors should provide better images with incorporated scale bars. Do the pattern of cell clustering in these images represent the stem-like (undifferentiated) and differentiated states of the tumor cells? Can the authors elaborate on this, as the top image shows ‘cluster or colony’ and not the middle or the bottom image?

We enhanced the image to have a sharper image, and included the scale of magnification, however, to accommodate all the panels we needed to decrease the size of the image. We apologize for this. The pictures were taken by the inverted microscope after seeding the same number of tumors and observing the growth rate of the cells. The clustering is because these tumors have significantly decreased growth rate and are not as healthy as what we see with tumors dissociated from tumor alone implanted mice. Indeed, when we analyzed the surface receptor expression these tumors expressed higher levels of the differentiation markers (MHC class I, CD54 and B7H1), and therefore, they are of differentiated phenotype. It is possible increased CD54 could be causing this clumping effect.

Figure 4H. Numbers within the panels and labels outside the panels are difficult to read. Figure H and I are not convincing.

We have sharpened the image, however, because of the small size of the panel to accommodate within figure 4, the numbers appear small. To alleviate the problem we have now presented a larger image in the supplemental file (Fig. S3C) to be able to read the numbers easier. In addition, we have determined the proportions of each subset in the CD45+ cells in the panel presented in S3C (S3D). Also the two independent assessment of CD45+ cells shown in the supplemental file (S3B and S3C) in the dissociated pancreatic cells we observe higher percentages of CD45+ cells in NK injected tumor bearing mice and NK-differentiated MP2 implanted tumors as compared to tumor alone implanted mice. We hope this has alleviated the concern.  

Figure 5. Change panel labels. Do panels J and K require labels ‘BM’?

We have done as suggested and removed all the references to the cell types from the panel

References #44 and #48 are duplicates. Please check all the references.

We have corrected as suggested